

# Reactive quenching of electronically excited $NO_2^*$ and $NO_3^*$ by $H_2O$ as potential sources of atmospheric HO*x* radicals

Terry J. Dillon[1,2] and John N. Crowley[1]

[1]Max Planck Institute for Chemistry, Division of Atmospheric Chemistry, Mainz, Germany
[2]Now at: Wolfson Atmospheric Chemistry Laboratories, Department of Chemistry, University of York. U.K

*Correspondence to*: John N. Crowley (john.crowley@mpic.de)

**Abstract.** Pulsed laser excitation of $NO_2$ (532 – 647 nm) or $NO_3$ (623 - 662 nm) in the presence of $H_2O$ was used to initiate the gas-phase reactions $NO_2^* + H_2O \rightarrow$ products (R5) and $NO_3^* + H_2O \rightarrow$ products (R12). No evidence for OH production in (R5) or (R12) was observed and upper-limits for OH production of $k_{5b} / k_5 < 1 \times 10^{-5}$ and $k_{12b} / k_{12} < 0.03$ were assigned. The

upper limit for $k_{5b} / k_5$ renders this reaction insignificant as a source of OH in the atmosphere and extends the studies (Crowley and Carl, 1997; Carr et al., 2009; Amedro et al., 2011) which demonstrate that the previously reported large OH yield by (Li et al., 2008) was erroneous. The upper limit obtained for $k_{12b} / k_{12}$ indicates that non-reactive energy transfer is the dominant mechanism for (R12), though generation of small but significant amounts of atmospheric HO*x* and HONO cannot be ruled out. In the course of this work, rate coefficients for overall removal of $NO_3^*$ by $N_2$ (R10) and by $H_2O$ (R12)

were determined: $k_{10} = (2.1 \pm 0.1) \times 10^{-11}$ cm$^3$ molecule$^{-1}$ s$^{-1}$ and $k_{12} = (1.6 \pm 0.3) \times 10^{-10}$ cm$^3$ molecule$^{-1}$ s$^{-1}$ which is more than a factor of three smaller than one previously reported value.

## 1 Introduction

The capacity of the atmosphere to oxidise trace gases released at the Earth's surface is sensitively dependent on the concentration of the hydroxyl radical, OH (Lelieveld et al., 2008). Most atmospheric OH is believed to be generated via a

combination of primary photolytic processes involving e.g. $O_3$ ($\lambda \leq 370$ nm) (R1, R2) and HONO ($\lambda$: 280-370 nm) as well as in reaction of NO with $HO_2$, the latter being formed in the troposphere via the oxidative degradation of organic trace gases.

| | | | |
|---|---|---|---|
| $O_3 + h\nu$ | $\rightarrow$ | $O(^1D) + O_2$ | (R1) |
| $O(^1D) + H_2O$ | $\rightarrow$ | 2 OH | (R2) |
| $HONO + h\nu$ | $\rightarrow$ | OH + NO | (R3) |

As a large fraction of the oxidation of organic trace gases is initiated by reaction with OH, the conversion of $HO_2$ back to OH (e.g. via reaction with NO) is often referred to as recycling; the relative importance of direct OH formation and recycling depending on the concentrations of organics and NO. Together, OH and $HO_2$ are referred to as HO*x*.

Any reaction that can generate OH or $HO_2$ directly or indirectly (e.g. via generation of a short lived OH-precursor such as HONO), thus contributes to atmospheric oxidation capacity. Processes that form HONO (both gas-phase and heterogeneous)



are therefore of great interest to atmospheric science and have been the subject of many studies (see e.g. (Stemmler et al., 2007; Li et al., 2014; Meusel et al., 2016). Two processes that may potentially generate HO$x$ and HONO are the gas-phase reactions of $H_2O$ with electronically excited nitrogen dioxide ($NO_2$ $A$ $^2B_2$ henceforth $NO_2^*$) and electronically excited nitrate radical ($NO_3$ $A$ $^2E''$ and $B$ $^2E'$, henceforth $NO_3^*$).

**1.1 $NO_2^*$ + $H_2O$**

The potential for this reaction to generate both OH and HONO was first discussed and evaluated by Crowley and Carl (1997) who highlighted a possible role in increasing OH production rates in the weakly illuminated winter troposphere. It was argued that non-dissociative absorption by $NO_2$ (R4) could lead to formation of OH and HONO in a process (R5b) that is exothermic for excitation wavelengths across visible absorption spectrum of $NO_2$ which extends to ≈ 650 nm.

$NO_2$ + $h\nu$ (≤ 650 nm)   →   $NO_2^*$   (R4)

$NO_2^*$ + $H_2O$   →   $NO_2$ + $H_2O$   (R5a)

$NO_2^*$ + $H_2O$   →   OH + HONO   (R5b)

$NO_2^*$ + $N_2$ / $O_2$   →   $NO_2$ + $N_2$ / $O_2$   (R6)

The rate of OH-formation following $NO_2$ excitation in the atmosphere depends on the OH yield ($k_{5b}$ / $k_5$) and on the relative
rates of $NO_2^*$ deactivation by $H_2O$ (R5) and by $N_2$ and $O_2$ (R6).  For details of the $NO_2$ cross-sections, quantum yields, quenching rate constants and associated photo-physics for these processes we refer to our previous publication (Crowley and Carl, 1997).

Crowley and Carl (1997), used 532 nm pulsed-laser excitation of $NO_2$, to determine an upper limit to the OH-yield of ($k_{5b}$ / $k_5$) ≤ 7 × 10$^{-5}$. Crowley and Carl (1997) also identified routes to O($^1$D) at shorter wavelengths that involved two-photon
excitation of $NO_2$, and which lead indirectly to OH formation via reaction of O($^1$D) with $H_2O$. Whilst of some utility in the laboratory, such processes that require multi-photon excitation are generally of no consequence for the atmosphere.

More than ten years later, Li et al. (2008) carried out similar experiments but at longer wavelengths (560 – 640 nm) and reached very different conclusions, deriving a yield of OH (and thus also HONO) close to 1 × 10$^{-3}$, a factor of 14 times larger than the upper limit of Crowley and Carl (1997). Calculations of the impact of (R4-R5) using the large yield reported by Li
et al. (2008) led to the conclusion that (R5) is important for air-quality under highly polluted conditions; use of the lower yield from Crowley and Carl (1997) resulted in minimal impact (Wennberg and Dabdub, 2008; Ensberg et al., 2010). Subsequent to the work of Li et al. (2008), two further experimental studies (Carr et al., 2009; Amedro et al., 2011) appeared to confirm the conclusions of Crowley and Carl (1997), and suggested that the high yield reported by (Li et al., 2008) was an experimental artefact, resulting from multi-photon laser-excitation of $NO_2$ in their focussed laser-beam (Amedro et al.,
2011). However, the experiments of Amedro et al. (2011) at 565 nm and Carr et al. (2009) at 563.5 and 567.5 nm used $NO_2^*$ prepared at wavelengths that covered only a small portion of the 560 – 630 nm  range from Li et al. (2008). The single wavelength (532 nm) used by Crowley and Carl (1997), whilst interrogating the same excited states of $NO_2$, was outside of



the range of wavelengths covered by Li et al. (2008). The principal goal of the experiments on (R5) described in this work was therefore to measure OH yields ($k_{5b}$ / $k_5$) using a range of photoexcitation wavelengths similar to those employed by Li et al. (2008) but avoiding potential complications related to multi-photon excitation.

## 1.2 $NO_3^*$ + $H_2O$

The $NO_3$ radical is generated throughout the atmospheric diel cycle via the oxidation of $NO_2$ by $O_3$:

$$NO_2 + O_3 \rightarrow NO_3 + O_2 \tag{R7}$$

At night, $NO_3$ can acquire mixing ratios of 100s of pptv. The high-reactivity of $NO_3$ towards unsaturated, organic trace gases (especially biogenically emitted ones in forested regions (Liebmann et al., 2018a; Liebmann et al., 2018b)) make it an important nocturnal oxidant. $NO_3$ is generally considered to be unimportant during daytime due to rapid photolysis. Rapid

photodissociation (R8a & R8b) following absorption of visible light, reduces the daytime $NO_3$ lifetime to only a few seconds, and usually limits mixing ratios to less than 1 pptv.

$$NO_3 + h\nu \rightarrow NO_2 + O(^3P) \tag{R8a}$$

$$NO_3 + h\nu \rightarrow NO + O_2 \tag{R8b}$$

$NO_3$ photo-physics has been the subject of many studies, up to 1991 reviewed by (Wayne et al., 1991). Briefly, the $NO_3$

absorption spectrum ($\approx$ 400 – 665 nm) is broad and diffuse with an extended excited-state lifetime of several hundred μs (Nelson et al., 1983) for excitation beyond the photo-dissociation limit. The extended lifetime results from coupling between ro-vibrational levels of the ground ($X\ ^2A_2$) state and the excited ($A\ ^2E''$ and $B\ ^2E'$) electronic states, so that excitation into the strongest feature (centred at $\approx$ 662 nm) can be considered to populate a manifold of mixed ground and excited electronic states (Carter et al., 1996). For simplicity, we refer to excited state $NO_3$ as $NO_3^*$.

$NO_3^*$ can dissociate (R8, dominant at excitation wavelengths < 630 nm), fluoresce (R9) and return to the ground-state or be quenched in collisions with the main atmospheric bath gases $N_2$, $O_2$ and $H_2O$ (R10-R12). Fluorescence and collisional quenching are important only at wavelengths longer than $\approx$ 630 nm.

$$NO_3^* \rightarrow NO_3 + h\nu \tag{R9}$$

$$NO_3^* + N_2 \rightarrow NO_3 + N_2^{\#} \tag{R10}$$

$$NO_3^* + O_2 \rightarrow NO_3 + O_2^{\#} \tag{R11}$$

$$NO_3^* + H_2O \rightarrow NO_3 + H_2O^{\#} \tag{R12a}$$

$$\rightarrow OH + HNO_3 \tag{R12b}$$

$$\rightarrow HO_2 + HONO \tag{R12c}$$

where # denotes formation of vibrationally hot products following energy transfer from $NO_3^*$. A simple analysis

demonstrates that formation of atmospheric free-radicals is thermodynamically feasible. Absorption of a 662 nm photon (the wavelength of maximum absorption by $NO_3$, see Figure 1), provides an excitation energy of 255 kJ mol$^{-1}$. Subsequent




formation of radical products from $NO_3^*$ is then exothermic: by 110 kJ mol$^{-1}$ for OH + HNO$_3$ (R12b); by 81 kJ mol$^{-1}$ for HO$_2$ + HONO (R12c).

The net result of NO$_3$ formation in (R7) and photolysis via the main channel, (R8a), is no change in NO$_X$ (NO$_X$ = NO + NO$_2$) or O$_3$. The net effect of formation in (R7) and photolysis via the minor (20%) channel (R8b) is conversion of NO$_2$ to NO (i.e.

no net loss of NO$_X$) and conversion of O$_3$ to O$_2$ (loss of odd-oxygen). Reaction of $NO_3^*$ with H$_2$O to form OH + HNO$_3$ (R12b) changes this picture dramatically. As illustrated in Figure 2, if $NO_3^*$ reacts with H$_2$O to form OH + HNO$_3$ (R12b), the net effect is conversion of NO$_2$ to HNO$_3$ (i.e. loss of NO$_X$) and conversion of O$_3$ and H$_2$O to OH. This process (R7, R12b) therefore allows formation of atmospheric OH from O$_3$ in the absence of actinic UV radiation normally required to generate O($^1$D) from O$_3$ (R1). If $NO_3^*$ reacts with H$_2$O to form OH + HONO + O$_2$, as in (R12c), the net effect is conversion of NO$_2$ to

NO (no loss of NO$_X$) and formation of two HO$x$ molecules, again bypassing the need for the actinic radiation in the UV. Using literature values for the wavelength dependent NO$_3$ absorption cross-sections (Yokelson et al., 1994) and photolysis quantum yields (Orlando et al., 1993) as well as actinic flux (calculated for 50 °N at two solar zenith angles, TUV) we calculate that, on average, 60% of actinic photons absorbed result in dissociation of NO$_3$. The residual 40% results in formation of $NO_3^*$ which can then undergo chemical and photo-physical transformation. Figure 1 gives an example of the

relative rates of photo-dissociation and (non-dissociative) photo-excitation across the NO$_3$ absorption spectrum.

The relative importance of fluorescence and the collisional deactivation processes depends on the fluorescence lifetime and the rate constants for quenching. Nelson et al. (1983) report two components to the NO$_3$ fluorescence decay they observed following excitation at 661.9 nm, with collision-free fluorescence-lifetimes of 27 and 340 μs.

The longer lived component (accounting for > 85% of the total fluorescence) was quenched by N$_2$ and O$_2$ with rate

coefficients of $k_{10}$ = (1.7 ± 0.2) × 10$^{-11}$ cm$^3$ molecule$^{-1}$ s$^{-1}$ and $k_{11}$ = (2.1 ± 0.02) × 10$^{-11}$ cm$^3$ molecule$^{-1}$ s$^{-1}$, respectively. Nelson et al. (1983) did not report a quenching rate coefficient for H$_2$O, but determined large quenching coefficients for propane (1.09 × 10$^{-10}$ cm$^3$ molecule$^{-1}$ s$^{-1}$) and nitric acid (3.07 × 10$^{-10}$ cm$^3$ molecule$^{-1}$ s$^{-1}$), presumably resulting from more efficient energy transfer due to higher densities of states in these polyatomics. A substantially larger rate coefficient for quenching of $NO_3^*$ by H$_2$O of $k_{12}$ = (6.9 ± 0.5) × 10$^{-10}$ cm$^3$ molecule$^{-1}$ s$^{-1}$) was reported by Fenter and Rossi (1997). The

quenching rate constants are sufficiently large that, at the pressures of N$_2$, O$_2$ and H$_2$O available in the troposphere, relaxation of $NO_3^*$ via fluorescence can be neglected.

The fraction, $f_{H2O}$, of tropospheric $NO_3^*$ that will be quenched by collision with H$_2$O rather than N$_2$ or O$_2$ is given by expression (1):

$$f_{H2O} = k_{12}[H_2O] / (k_{12}[H_2O] + k_{10}[N_2] + k_{11}[O_2]) \qquad (1)$$

Using this expression we calculate that, at the Earth's surface (1 bar pressure) and a temperature of 25 °C, $f_{H2O}$ can vary between 0.2 and 0.5 for relative humidities between 20 and 80%. As mentioned above, daytime concentrations of NO$_3$ are generally low due to rapid photolysis (and reaction with NO) though measurements in polluted environments indicate



maximum daytime concentrations of $[NO_3] \approx 1 \times 10^8$ molecule $cm^{-3}$ (Geyer et al., 2003). The atmospheric production rate of OH via $NO_3$ excitation may be written:

$$P_{OH}(NO_3^*) = J_{exci}[NO_3]\, f_{H2O} \tag{2}$$

Using an $NO_3$ concentration $1 \times 10^8$ molecule $cm^{-3}$ and $J_{exci} = 0.15$ $s^{-1}$ (Figure 1) enables us to calculation an OH production rate (at 80% relative humidity) of $7.5 \times 10^6$ molecule $cm^{-3}$ $s^{-1}$ if all quenching of $NO_3^*$ by $H_2O$ is reactive and forms OH. To put this value in context, we note that typical OH production rates from photolysis of $O_3$ are around $2 \times 10^5$ molecule $cm^{-3}$ $s^{-1}$, a factor of $\approx 40$ lower. The principal objective of this work was therefore to determine the OH production rate via $NO_3$ photoexcitation and subsequent reaction of $NO_3^*$ with $H_2O$ (R12b). To best constrain these measurements, rate coefficients for total removal (quenching and chemical reaction) of $NO_3^*$ by $H_2O$ ($k_{12}$) and $N_2$ ($k_{10}$) were determined.

## 2 Experimental

All experiments were conducted in a 500 $cm^3$ jacketed photolysis cell as described previously (Wollenhaupt et al., 2000; Dillon et al., 2006). Laser light entered and exited the reaction vessel via Brewster-angle quartz-windows; laser-fluence at each wavelength being recorded using a Joule-meter located behind the exit-window. An excimer laser was used to generated $\approx 20$ ns pulses of light at 193 nm (ArF) or 248 nm (KrF). Dye-lasers pumped by Nd-YAG lasers were used to generate pulsed ($\approx 6$ ns) tuneable radiation at visible wavelengths.

The pressure and the gas flow rate (300 - 2000 $cm^3$ (STP) $min^{-1}$) were regulated to ensure that a fresh gas sample was available for each laser pulse for operation at 10 Hz. The pulsed laser-based schemes for generation of excited $NO_2$ and $NO_3$ are described below, as are the schemes for calibration of the OH signal.

Concentrations of the key reactants and precursors ($NO_2$, $HNO_3$ and $H_2O$) were monitored by UV-vis. absorption spectroscopy, reducing potential uncertainties in each of these parameters to $\leq 10\%$. $NO_2$ was measured in-situ using a multi-pass absorption cell positioned upstream of the reactor. Light from a halogen lamp passing through the cell was focused onto the entrance slit of a 0.5 m monochromator. A diode-array detector was used to record $NO_2$ absorption in the visible range of light between $398 \leq \lambda \leq 480$ nm at an instrumental resolution of 0.32 nm, determined from the full width at half maximum (FWHM) of the 436.8 nm Hg emission line. Optical absorption by $HNO_3$ and $H_2O$ was determined using a "dual beam" absorption cell (184.95 nm, $l$ = 43.8 nm) located downstream of the photolysis reactor. $NO_2$ concentrations were calculated using a literature reference spectrum (Vandaele et al., 1998). Concentrations of $HNO_3$ and $H_2O$ were calculated using cross-sections of $1.61 \times 10^{-17}$ $cm^2$ $molecule^{-1}$ (Dulitz et al., 2018) and $7.22 \times 10^{-20}$ $cm^2$ $molecule^{-1}$ (Creasey et al., 2000).

The output from a Nd-YAG pumped dye-laser operating with Rhodamine 6G dye was frequency doubled to 282 nm and used to detect OH via excitation the $A^2\Sigma$ (v = 1) $\leftarrow$ $X^2\Pi$ (v = 0) transition close to 282 nm. Laser-induced fluorescence was detected by a photomultiplier tube shielded by a combination of a 309 nm ($\pm$ 5 nm) interference filter and BG 26 (glass)



filter. Directly following experiments to measure formation of OH in the title reactions, known amounts of OH were generated via pulsed laser photolysis of $HNO_3$ (at 248 nm) or $H_2O$ (at 193 nm).

$HNO_3 + h\nu$ (248 nm) $\rightarrow$ $OH + NO_2$          (R13)

$H_2O + h\nu$ (193 nm) $\rightarrow$ $OH + H$          (R14)

For the experiments on $NO_2^*$, a small flow of $HNO_3$ diluted in $N_2$ was added to the $N_2$ bath and a series of experiments was conducted that covered a range of laser fluences at 248 nm. The additional flow was compensated by reducing the main $N_2$ flow so that different concentrations of OH were generated in essentially unchanged conditions of pressure, temperature [$NO_2$] and [$H_2O$]. When using (R14) to calibrate the OH signal, the $NO_2$ supply to the experiment was replaced with $N_2$, and 193 nm light used to dissociate OH from the $H_2O$ already present (in unchanged conditions of pressure, temperature and

[$H_2O$]).

The uncertainty associated with conversion of LIF signals into OH concentrations stemmed partially from uncertainties in (measured) [$HNO_3$] and [$H_2O$] but was dominated by uncertainty in the measurement of the laser fluence at the centre of the reactor. Such measurements depended on both the accuracy of the Joule-meter and corrections for beam divergence and the assumption of a homogeneous light intensity over the cross-section of the laser beam. An overall uncertainty of 40% was

estimated for the conversion of LIF-signals to absolute [OH] required for ($k_{5b}$ / $k_5$) determinations. For determination of $k_{12b}$ / $k_{12}$, the self-calibrating chemistry (R13, R15) results in a smaller contribution of laser fluence uncertainty to the overall uncertainty which is dominated by assumptions regarding the $NO_3$ profile (see later).

Chemicals: $NO_2$ (ABCR 99.99%) was subject to repeated freeze-pump-thaw cycles at 77 K prior to dilution in $N_2$ and storage in blackened glass bulbs; $H_2O$ ("milli-Q" de-ionised water) and $HNO_3$ (prepared in house from $H_2SO_4$ + $KNO_3$) were

added to the reactor via bubblers; $O_3$ was generated via electric discharge through $O_2$ in a commercial ozoniser (Anseros); $N_2O_5$ was prepared by mixing $O_3$ with $NO_2$ and trapping the resulting $N_2O_5$ at 195 K (Wagner et al., 2008); $N_2$ and $O_2$ (Westfalen, 99.999%) were used as supplied.

## 3 Results and Discussion

### 3.1 $NO_2^*$ + $H_2O$ (R5)

A Nd-YAG pumped dye-laser was used to generate 532 and 567 - 647 nm light for pulsed laser excitation of $NO_2$. Reagent concentrations and conditions for these experiments are given in Table 1. In general, large concentrations of $H_2O$ were used to promote reaction of $NO_2^*$ over deactivation by other colliders, notably $N_2$, and to ensure that changes in other reagent concentrations (e.g. for calibration, see above) had a minimal effect on fluorescence quenching or other processes that impact on OH-LIF detection sensitivity.

Figure 3 displays the results of an experiment in which $NO_2$ was excited at 532 nm (at $t$ = 280 μs) to generate $10^{13}$ to $10^{14}$ molecule $cm^{-3}$ of $NO_2^*$. The delay of 280 μs is the time between the triggering of the flash-lamps (at $t$ = 0) and the Q-switch



of the YAG-laser. The solid black triangles were obtained with the OH-excitation laser tuned to 282 nm (on resonance) and indicate a change in signal ≈ 200 to 350 μs. This signal does not display the kinetic behaviour of OH in this chemical environment and remains when the OH-excitation laser is tuned off resonance (red triangles). It is also present when the 532 nm light is blocked and we conclude that this weak signal, having neither kinetics or spectroscopy characteristic of OH, is an

artefact with electronic origin, possibly related to the output of the pulse generator used to trigger the laser Q-switch.

The data represented by open circles (roughly independent of reaction time) are the results of OH-calibration experiments using the 193 nm photolysis of $H_2O$ ($1.5 \times 10^{17}$ molecule $cm^{-3}$) at four laser fluences between 0.3 and 6.8 mJ $cm^{-2}$ in the absence of $NO_2$. The roughly constant OH level over 1000 μs is consistent with the fact that OH does not react with any components of the gas-mixture. An experiment at 193 nm using the same OH-generation scheme but in the presence of $NO_2$

is displayed as solid stars. OH now decays exponentially at a rate which is consistent with its loss via reaction with $NO_2$. In this experiment, some OH was also generated by the reaction of $O(^1D)$ (formed by the 193 nm photolysis of $NO_2$) with $H_2O$ and it was not used for calibration purposes. The signals obtained in the absence of $NO_2$ were converted to OH concentrations (right y-axis) using Joule meter readings as described in section 2.1.

The solid black line in Figure 3 represents the OH-signal and concentration expected from our experimental conditions ($NO_2$

concentration, $H_2O$ concentration, total pressure and 532 nm laser fluence) and literature data for $NO_2$ absorption cross-sections, $NO_2^*$ deactivation rate constants and the yield of OH from $NO_2^*$ + $H_2O$ reported by Li et al. (2008). From this plot, it is immediately apparent, that our data are not consistent with the large yield of OH reported by Li et al. (2008). In order to rule out the possibility that this is a result of using different excitation wavelengths, similar experiments were carried out in which we explored different regions of the $NO_2$ absorption spectrum. OH signals were not observed at any wavelength,

enabling us to set upper limits to $k_{5b} / k_5$. The upper limits were calculated from the minimum observable OH-signal (assumed to be twice the RMS noise levels on the OH-signal) and accounting for uncertainty in parameters such as laser fluence (30%), $NO_2$ concentration (10%) and concentration of $H_2O$ (10%).

The results are summarised in Table 1 which lists the experimental conditions in detail and in Figure 4 where we also compare to literature determinations of $k_{5b} / k_5$. The present dataset and those reported by Crowley and Carl (1997), Amedro

et al. (2011) and Carr et al. (2009) found OH formation in the reaction between $NO_2^*$ and $H_2O$ to be inefficient, with upper limits to $k_{5b} / k_5$ of between $6 \times 10^{-6}$ and $1.4 \times 10^{-4}$ at all wavelengths investigated. Together, these datasets contradict the yield of $1 \times 10^{-3}$ reported by Li et al (2008) for excitation across the wavelength range 560 to 630 nm. Our dataset, covering three absorption features of the $NO_2$ absorption spectrum within the range reported by Li et al. (2008) also rules out that the poor agreement is due to use of different excitation wavelengths. As discussed by Amedro et al. (2011) the use of focussed

laser beams and resulting multi-photon processes are the most likely explanation for Oh formation in the work of Li et al (2008). The results from this work reduce the maximum yield of OH from the reaction of $NO_2^*$ with $H_2O$ to $6 \times 10^{-6}$ at 532 nm as opposed to $7 \times 10^{-5}$ measured by Crowley and Carl (1997). The assumption that this value is valid across the non-




dissociative part of the absorption spectrum of $NO_2$, enables us to conclude that formation of atmospheric OH (and HONO) via R5b is insignificant.

### 3.2 $NO_3^*$ + $H_2O$ (R12)

#### 3.2.1 Generation of $NO_3$

For the experiments to investigate the reaction of $NO_3^*$ with $H_2O$ (R12), $NO_3$ was generated via the reaction of OH with known amounts of $HNO_3$ (R15).

OH + $HNO_3$ $\rightarrow$ $NO_3$ + $H_2O$ (R15)

The rate constant and the yield of $NO_3$ (unity) for (R15) are well known (Brown et al., 1999; Brown et al., 2001; Carl et al., 2001; Dulitz et al., 2018) enabling the time-dependent $NO_3$ concentration profile to be calculated if the initial amount of OH
is known. This initial concentration of OH depends on the 248 nm laser-fluence (measured by Joule-meter, uncertainty 30%) and the $HNO_3$ concentration (measured by optical absorption at 185 nm, uncertainty 10%). As OH was formed from $HNO_3$ photolysis (R13), and the OH decay monitored, these experiments were self-calibrating as long as a sufficient excess of $[HNO_3] >> [OH] \approx [NO_3]$ was maintained. In the conditions employed in this work (see Table 2), radical loses via unwanted self- and cross-reactions of OH and $NO_3$ were < 5% of the total OH loss rate which was dominated by (R15). In experiments
to measure the rate constant for $NO_3^*$ quenching by $N_2$ (R10) and $H_2O$ (R12), $NO_3$ was generated via the 248 nm photolysis of $N_2O_5$:

$N_2O_5$ + $h\nu$ (248 nm) $\rightarrow$ $NO_3$ + $NO_2$ (R16)

In this scheme of $NO_3$ generation $NO_3$ is formed instantaneously (in contrast to the reactions (R13 and R14).

#### 3.2.2 Quenching of $NO_3^*$ by $N_2$ and $H_2O$ ($k_{10}$ and $k_{12}$)

The fate of electronically excited $NO_3$ radicals in the atmosphere is controlled by the relative rate of quenching by $H_2O$ and the predominant bath-gases $N_2$ and $O_2$, which depends both on the concentration of $H_2O$ and on the quenching rate coefficients $k_{10}$, $k_{11}$ and $k_{12}$. As the rate constant for quenching of $NO_3^*$ by $H_2O$ ($k_{12}$) has been addressed only briefly in a single study (Fenter and Rossi, 1997) and the value derived ($6.9 \times 10^{-10}$ cm$^3$ molecule$^{-1}$ s$^{-1}$) is unexpectedly large, we chose to re-measure $k_{12}$. In these experiments, $NO_3$ was generated in (R16) and He was used as the main bath gas, with traces of $N_2$
and $H_2O$ added.

An excitation laser-pulse at 662 nm was triggered when the $NO_3$ concentration was close to its maximum value (i.e. when > 95% of the primary-OH had been consumed by reaction with $HNO_3$) to generate $NO_3^*$. Time dependent fluorescence from $NO_3^*$ ($\lambda$ > 690 nm) was detected using a red-sensitive photo-multiplier and recorded on a 100 MHz, digital oscilloscope. Fluorescence decay constants in the presence of various concentrations of $H_2O$ were then used to derive $k_{12}$. We also
conducted a set of experiments using $N_2$ as quenching molecule to test our experimental methodology by comparison with literature measurements of $k_{10}$.



NO$_3$ fluorescence profiles from these experiments are displayed in Figure 5, where datasets are depicted in which various amounts of N$_2$ (Fig. 5a) and H$_2$O (Fig. 5b) were added to the He bath gas. The fluorescence decay rate constant ($k_f'$) derives from the sum of processes that depopulate the excite state and includes fluorescence, inter-system crossing as well as quenching by N$_2$, H$_2$O and N$_2$O$_5$ with rate constants ($k_f$), $k_{ISC}$, $k_f(N_2)$, $k_f(H_2O)$ and $k_f(N_2O_5)$, respectively.

$k_f' = k_f + k_{ISC} + k_q(N_2)[N_2] + k_q(H_2O)[H_2O] + k_q(N_2O_5)[N_2O_5]$

In line with previous studies Nelson et al. (1983), the slow component of the NO$_3$ fluorescence was found to decay mono-exponentially (black and red lines in Figs. 5a and 5b) and depended on the pressure of N$_2$ or H$_2$O.

The decay constant ($k_f'$) was derived from exponential fits to the data and plotted against the concentration of N$_2$ or H$_2$O (Figure 6) to obtain (from the slopes) the rate constants $k_{10}$ and $k_{12}$ for quenching by N$_2$ and H$_2$O, respectively. Assuming

negligible contribution from OH, NO$_2$ and NO$_3$, due to their low concentrations, the y-axis intercepts in Fig. 6 ($\approx$ 0.5-0.8 ×10$^6$ s$^{-1}$) are the combined terms $k_f + k_{ISC} + k_q(N_2O_5)[N_2O_5]$, where $k_q(N_2O_5)$ is the unknown rate constant for quenching of NO$_3$* by N$_2$O$_5$. As the collision-free lifetime of excited NO$_3$ is several hundred μs, the terms $k_f$ and $k_{ISC}$ contribute insignificantly to the fluorescence decay. The intercept ($\approx$ 5-8 × 10$^5$ s$^{-1}$) is consistent with N$_2$O$_5$ concentrations in the range 10$^{15}$ molecule cm$^{-3}$ and a value of $k_q(N_2O_5)$ of the order of 10$^{-10}$ cm$^3$ molecule$^{-1}$ s$^{-1}$.

Our result obtained in N$_2$, $k_{10} = (2.1 \pm 0.2) \times 10^{-11}$ cm$^3$ molecule$^{-1}$ s$^{-1}$, is in reasonable agreement with the value of $(1.7 \pm 0.2) \times 10^{-11}$ cm$^3$ molecule$^{-1}$ s$^{-1}$ reported by Nelson et al. (1983). In contrast, our result for quenching by water vapour, $k_{12} = (1.6 \pm 0.3) \times 10^{-11}$ cm$^3$ molecule$^{-1}$ s$^{-1}$, more than a factor of three lower than that reported by Fenter and Rossi (1997). As both studies used 662 nm excitation of NO$_3$ and similar methods to derive $k_{12}$, the differences are likely to be related to the measurement of the H$_2$O concentration. As we measured the H$_2$O concentration in situ (optical absorption at 185 nm) the

uncertainty of our result is expected to be determined by uncertainty in the absorption cross section of H$_2$O at this wavelength, which, based on good agreement across several measurements (Cantrell et al., 1997; Hofzumahaus et al., 1997; Creasey et al., 2000) we estimate to be < 10%. Fenter and Rossi (1997) relied on flow measurements to derive the concentration of H$_2$O in their experiments. Because of this, we consider our measurement of $k_{10}$ the more accurate and use this value for further evaluation of our experiments to derive $k_{12b} / k_{12}$.

### 3.2.3 Yield of OH from NO$_3$* + H$_2$O

Figure 7 displays the results of an experiment using three pulsed lasers. The first (excimer laser at time zero) generated OH from the 248 nm photolysis of HNO$_3$. In this particular experiment the HNO$_3$ concentration (monitored at 185 nm) was 6.3 × 10$^{15}$ molecule cm$^{-3}$ and a laser-fluence of 13 mJ cm$^{-2}$ was used to generate 2.0 × 10$^{12}$ OH cm$^{-3}$. This OH monitored by the 282 nm LIF-laser out to a reaction time of 10ms (open circles in Fig. 7), was observed to decay at a rate consistent with its

well-characterised reaction with HNO$_3$ $k_{15}$(298 K, 22.5 Torr) = 1.3 × 10$^{-13}$ cm$^3$ molecule$^{-1}$ s$^{-1}$ (Dulitz et al., 2018). NO$_3$ is the unique product of reaction (R15) (Brown et al., 2001; Carl et al., 2001). The NO$_3$ profile (dashed line), calculated from initial OH and HNO$_3$ concentrations, is also displayed in Fig. 7. Here we calculate that 97% of the initial OH will react with



$HNO_3$, the balance resulting from diffusion and reaction with $NO_2$. The decay of $NO_3$ at long reaction times is due to $NO_3$ diffusion from the reaction volume so that its concentration at 8.28 ms (when the 662 nm laser is triggered) was reduced by $\approx$ a factor of two compared to the stoichiometric yield of $2 \times 10^{12}$ molecule $cm^{-3}$ (i.e. when all OH is converted to $NO_3$). The decay of $NO_3$ was calculated from the known diffusion loss constant for OH at this pressure and the relative reduced-masses

of OH and $NO_3$. A delay of 8.28 ms allowed the primary-OH to decay to very low values (i.e. $\approx 10^9$ molecule $cm^{-3}$) before triggering the 662 nm excitation laser. The measured laser fluence at 662 nm was then combined with the $NO_3$ concentration at 8.28 ms to calculate the fractional excitation of $NO_3$ (generally about 10%) and thus the concentration of $NO_3^*$ formed. When using very large laser fluences at 662 nm we calculate that the transition was saturated and then assume equal concentrations of ground and excited state $NO_3$ directly after the excitation pulse.

The solid lines starting at $t = 8.28$ ms represent the expected OH-signal if the value of $k_{12b} / k_{12}$ were 0.0, 0.01, 0.05 and 0.1 and were calculated using the rate constants for quenching of $NO_3^*$ by $N_2$ and $H_2O$ as derived in this study as well as the concentrations of $N_2$ and $H_2O$.

Clearly, the data from the experiment illustrated in Fig. 7 is consistent with a value of $k_{12b} / k_{12}$ that lies between 0 and 1%. Similar experiments were repeated for different starting conditions and photoexcitation wavelengths (623, 629 and 662 nm)

corresponding to strong absorption features of $NO_3$. No evidence for OH production in (R12) was observed in any experiment and an upper limit to the yield of OH was obtained from the random noise on the experimental OH-trace data and the expected OH signal. These values are tabulated in Table 2. The major sources of uncertainty in the calculated OH-yield are uncertainty in the measurement of laser fluences (30%) required to calculate the initial OH and $NO_3^*$ concentrations and assumptions related to the (unmeasured) $NO_3$ time profile. $NO_3$ is relatively unreactive in this system as it does not react

with $HNO_3$ and only slowly with $NO_2$ (formed in R13) at these pressures. We calculate that $\approx$ 5% of the $NO_3$ formed is lost via reaction with OH ($k(OH+NO_3) = 2 \times 10^{-11}$ $cm^3$ $molecule^{-1}$ $s^{-1}$) (Atkinson et al., 2004), its major removal processes being diffusive transport. The diffusive loss rate constant for $NO_3$ in this system was calculated from the known diffusive loss rate constant of OH under the same conditions of pressure and temperature. In the absence of corroborative measurements of the $NO_3$ profile in these experiments, we conservatively assume a factor two uncertainty in the $NO_3$ concentration at the time of

the excitation pulse. We thus derive an upper-limit of $k_{12b} / k_{12} < 0.03$ following photoexcitation at 623, 629 and 662 nm. This indicates either that the rapid quenching of $NO_3^*$ by $H_2O$ predominantly involves energy transfer rather than reaction, or that the products formed in reactive quenching to not include OH.

## 4 Atmospheric implications and conclusions

The results obtained in this work and elsewhere (Crowley and Carl, 1997; Carr et al., 2009; Amedro et al., 2011) clearly

demonstrate that the large values of $k_{5b} / k_5$ reported by Li et al. (2008) were erroneous. In this work we were able to reproduce, extend and improve upon previous results (i.e. obtain smaller upper-limits for $k_{5b} / k_5$). The extension of the database to a wider range of photoexcitation wavelengths was important, since the majority of the data from Li et al. (2008)



were obtained at wavelengths red-shifted from those of the other groups. In the modelling study by Wennberg and Dabdub (2008) the largest impacts of (R5b) on air quality (enhancements in $O_3$ of ≈ 40%) were found when using $k_{5b} / k_5 = 10^{-3}$ from Li et al. (2008). Small but still significant impacts changes in $O_3$ and particle mixing ratios were calculated when using the upper-limit of $k_{5b} / k_5 = 7 \times 10^{-5}$ provided by (Crowley and Carl, 1997). Results from this work, with upper-limits to $k_{5b} / k_5$ an

order of magnitude smaller than those available previously, enable us to conclude that the formation of OH in $NO_2{}^* + H_2O$ is not an important atmospheric process.

Our upper limits of 3% to OH formation from the reactive quenching of $NO_3{}^*$ by $H_2O$ can be put in context using equations (1) and (2). We combine our measurements of $k_{10} = 2.1 \times 10^{-11}$ $cm^3$ $molecule^{-1}$ $s^{-1}$ and $k_{12} = 1.6 \times 10^{-10}$ $cm^3$ $molecule^{-1}$ $s^{-1}$ with the literature value for $k_{11}$ ($2.1 \times 10^{-11}$ $cm^3$ $molecule^{-1}$ $s^{-1}$, Nelson et al. (1983)) to derive $f_{H_2O} = 0.16$ at 25°C and a

relative humidity of 80%. Using the same excitation rates and concentrations of $NO_3$ described in section 1.1 and our upper limit of $k_{12b} / k_{12} = 0.03$, we derive an OH production rate of ≈ $7 \times 10^4$ OH $cm^{-3}$ $s^{-1}$. Whilst this value is ≈ two orders of magnitude lower than that calculated in section 1.1 where we assumed that all $NO_3{}^* + H_2O$ interactions form OH and used the high value of $k_{12}$ from the literature (Fenter and Rossi, 1997), it may still represent an important contribution to OH formation in some environments. The low yield of OH most likely results from the dominance of collisional energy transfer

over reactive quenching of $NO_3$ by $H_2O$ ($k_{12b} << k_{12}$).

However, we also consider the possibility that the non-observation of OH in our experiments reflects the fact that the preferred products are HONO + $HO_2$ (i.e. $k_{12c} > k_{12b}$) even though the molecular rearrangements required to form these products are less straightforward than for formation of OH and $HNO_3$ if excited state $NO_3$ has the same (approximate) $D_{3h}$ symmetry as the ground state and formally contains no O-O bonds. The conversion of $HO_2$ to (detectable) OH via addition of

NO was not feasible owing to the rapid reaction of NO with $NO_3$ ($k(HO_2 + NO)$ ≈ $8 \times 10^{-12}$ $cm^3$ $molecule^{-1}$ $s^{-1}$, $k(HO_2 + NO)$ ≈ $3 \times 10^{-11}$ $cm^3$ $molecule^{-1}$ $s^{-1}$ (Atkinson et al., 2004; IUPAC, 2018).

Given that our experiments were blind to formation of $HO_2$ or HONO a detailed discussion of the atmospheric role of reaction (R12c) is not warranted. However, the potential importance of reaction R12c can be illustrated by assuming a 10% yield of HONO and $HO_2$ ($k_{12c} / k_{12} = 0.1$) and the same temperature, $NO_3$ concentration and relative humidity outlined above.

With this scenario, we calculate production rates of $HO_2$ and HONO of ≈ $2 \times 10^5$ molecule $cm^{-3}$ $s^{-1}$. For HONO, this production rate is comparable to its formation in the gas-phase reaction between OH and NO under low $NO_X$ conditions but lower than the missing production rate of ≈ $1-5 \times 10^6$ molecule $cm^{-3}$ $s^{-1}$ that has been observed in several environments as summarised by Meusel et al. (2016). In terms of $HO_2$ formation, a rate of $2 \times 10^5$ molecule $cm^{-3}$ $s^{-1}$ would be comparable to that obtained by the photolysis of ≈ 0.5 ppbv of HCHO (assuming a J-value for HCHO of ≈ $2 \times 10^{-5}$ $s^{-1}$). In conclusion,

whilst our experiments indicate that the reactive quenching of excited $NO_3$ by water vapour is inefficient compared to collisional deactivation, we cannot rule out that this reaction plays a role in HO$x$ or HONO production. Experiments sensitive to $HO_2$ or HONO formation and theoretical calculations could help shed light on this.




**Acknowledgements**

We thank the Deutsche Forschungsgemeinschaft (DFG) for partial financial support of this research (CR 246/2-1).

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



**Table 1  Experimental conditions and results for $NO_2^*$ + $H_2O$ (R5)**

| $\lambda$ | $E_\lambda$ | $n$ | $P$ | [$H_2O$] | [$NO_2$] | OH calibration | $k_{5b}/k_5$ $(10^{-6})$ |
|---|---|---|---|---|---|---|---|
| 532 | 7.4 | 2 | 5 | 22 | 1.6 | (R13) | < 9 |
| 532 | 14.3 | 3 | 14 | 150 | 4.0 | (R13) | < 6 |
| 564.5 | 7.9 | 2 | 20 | 288 | 4.1 | (R14) | < 9 |
| 592.4 | 4.3 | 3 | 14 | 150 | 4.0 | (R13) | < 70 |
| 612.7 | 5.5 | 2 | 17 | 260 | 5.1 | (R14) | < 42 |
| 647.0 | 14.5 | 2 | 14 | 150 | 3.9 | (R14) | < 200 |
| 647.0 | 14.5 | 1 | 28 | 210 | 4.0 | (R14) | < 140 |

Notes: $\lambda$ = excitation wavelength (nm). $E_\lambda$ = excitation laser fluence (in $10^{16}$ photons pulse$^{-1}$ cm$^{-2}$); $n$ = number of repeat experiments; $P$ = bath-gas ($N_2$) pressure (mbar); units of concentration were $10^{15}$ molecule cm$^{-3}$.

**Table 2 – Experimental conditions and results for $NO_3^*$ + $H_2O$ (R12)**

| $\lambda$ | $E_\lambda$ | $P$ | [$H_2O$] | [$HNO_3$] | $k_{12b}/k_{12}$ |
|---|---|---|---|---|---|
| 623 | $6.2\times10^{16}$ | 34 | 70 | 6.3 | < 0.017 |
| 623 | $5.9\times10^{16}$ | 34 | 70 | 5.8 | < 0.015 |
| 629 | $5.5\times10^{16}$ | 34 | 70 | 5.8 | < 0.019 |
| 629 | $5.5\times10^{16}$ | 34 | 70 | 5.7 | < 0.025 |
| 662 | $15.4\times10^{17}$ | 33 | 45 | 6.3 | < 0.017 |
| 662 | $12.0\times10^{17}$ | 33 | 47 | 6.3 | < 0.003 |
| 662 | $1.5\times10^{16}$ | 16 | 49 | 8.0 | < 0.080 |
| 662 | $1.38\times10^{16}$ | 16 | 49 | 6.0 | < 0.012 |

Notes: $\lambda$ = excitation wavelength (nm). $E_\lambda$ = excitation laser fluence (in $10^{16}$ photons pulse$^{-1}$ cm$^{-2}$); $P$ = bath-gas ($N_2$) pressure (mbar); units of concentration were $10^{15}$ molecule cm$^{-3}$.




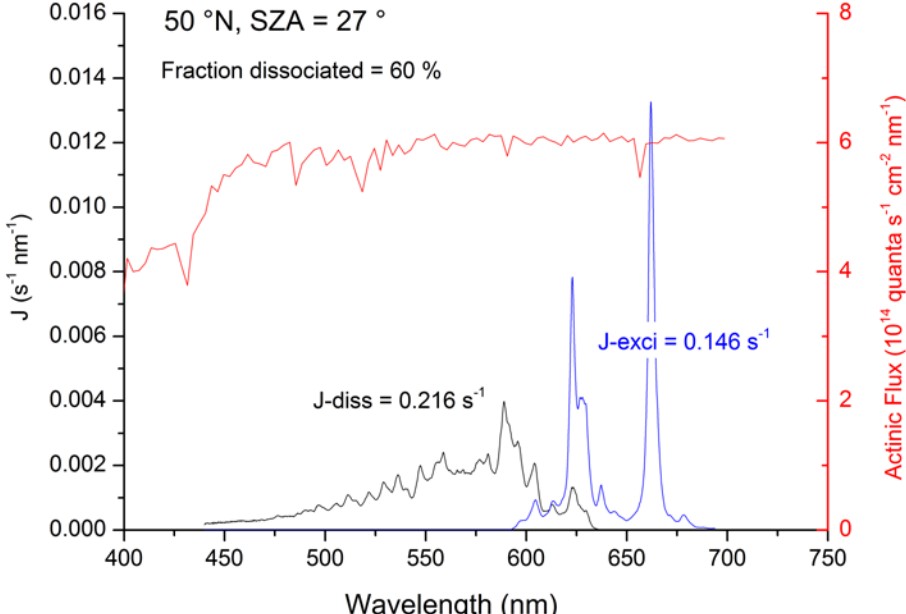

**Figure 1**: Rate constants for dissociative (black line, J-diss) and non-dissociative (blue line, J-exci) excitation of $NO_3$. The data use solar radiation actinic flux at the surface at 50°N and a solar zenith angle (SZA) of 27 ° (red line) as well as the $NO_3$ absorption cross sections and quantum yields. J-values (and fraction of $NO_3$ dissociated) were obtained by integration of the excitation rate (quanta $s^{-1}$ $nm^{-1}$) over the wavelength range of absorption.



| R4 | $NO_2 + h\nu$ | $\rightarrow$ | $NO_2^*$ |
| R5b | $NO_2^* + H_2O$ | $\rightarrow$ | $OH + HONO$ |
| R3 | $HONO + h\nu$ | $\rightarrow$ | $OH + NO$ |
| **Net:** | $\mathbf{NO_2 + H_2O}$ | $\rightarrow$ | $\mathbf{2\ OH + NO}$ |
| | | | |
| R7 | $NO_2 + O_3$ | $\rightarrow$ | $NO_3 + O_2$ |
| R8c | $NO_3 + h\nu$ | $\rightarrow$ | $NO_3^*$ |
| R12b | $NO_3^* + H_2O$ | $\rightarrow$ | $OH + HNO_3$ |
| **Net:** | $\mathbf{NO_2 + O_3 + H_2O \rightarrow OH + HNO_3 + O_2}$ | | |
| | | | |
| R7 | $NO_2 + O_3$ | $\rightarrow$ | $NO_3 + O_2$ |
| R8c | $NO_3 + h\nu$ | $\rightarrow$ | $NO_3^*$ |
| R12c | $NO_3^* + H_2O$ | $\rightarrow$ | $HO_2 + HONO$ |
| R3 | $HONO + h\nu$ | $\rightarrow$ | $OH + NO$ |
| **Net:** | $\mathbf{NO_2 + O_3 + H_2O \rightarrow OH + HO_2 + NO + O_2}$ | | |

**Figure 2**: Net effects of reactive removal of $NO_2^*$ and $NO_3^*$ by $H_2O$.





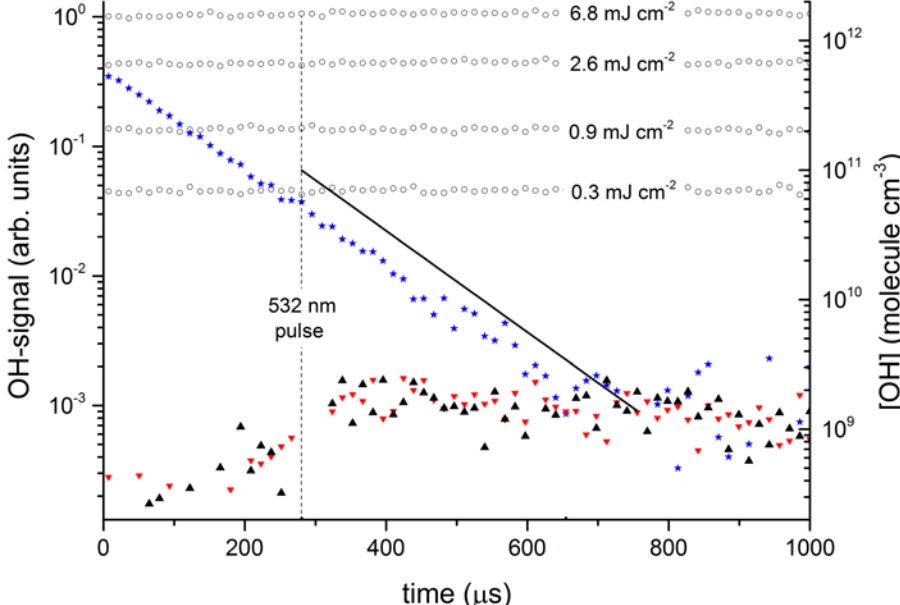

**Figure 3**: Photo-excitation of $NO_2$ at 532 nm. The open circles are OH-calibrations obtained by the 193 nm photolysis of $H_2O$ at different laser fluences (mJ cm$^{-2}$). The solid stars are data points from an OH calibration in the presence of $NO_2$. The black triangles are data obtained by photoexcitation of $[NO_2] = 4.0 \times 10^{15}$ molecule cm$^{-3}$ using 532 nm (50 mJ cm$^{-2}$) in the presence of $[H_2O] = 1.5 \times 10^{17}$ molecule cm$^{-3}$. The red triangles are the results of an identical experiment, but with the OH-excitation laser tuned away from the OH-feature at 282 nm. The solid black line represents the OH signal and concentration expected from the yield of OH from $NO_2^* + H_2O$ reported by Li et al. (2008).




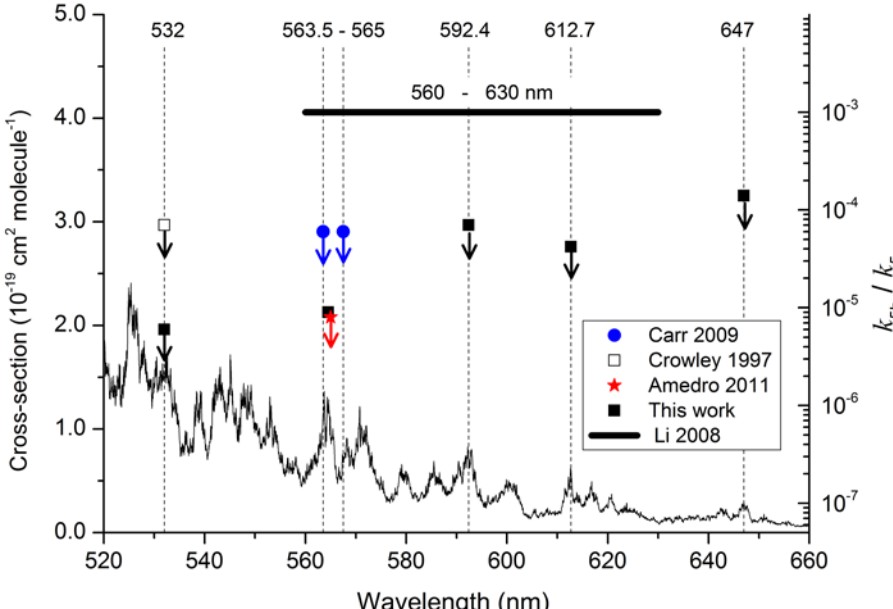

**Figure 4**: Summary of data obtained following photoexcitation of NO$_2$ at various wavelengths. The data from this study, Crowley and Carl (1997), Carr et al. (2009) and Amedro et al. (2011) are all upper limits, indicated by the down-arrows. The NO$_2$ absorption cross-sections were taken from Van Daele et al (1998).





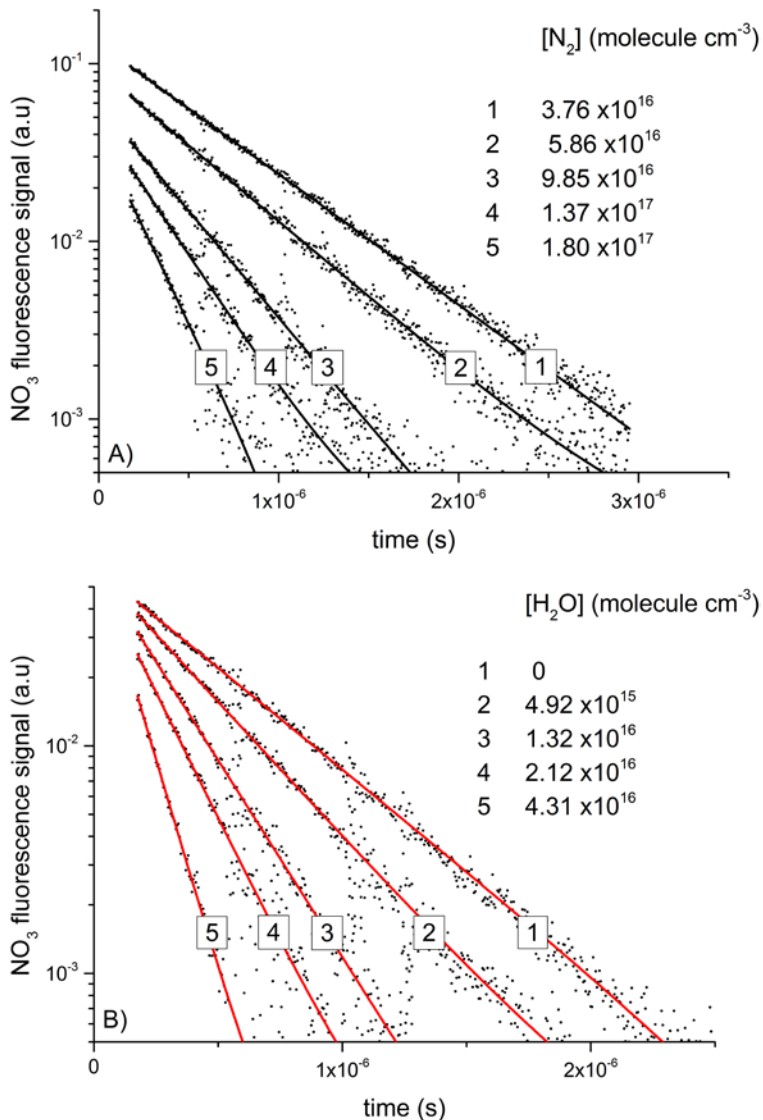

**Figure 5**: Exponential decay of fluorescence from $NO_3$ following photoexcitation at 623 nm in the presence of $N_2$ (panel A) and $H_2O$ (panel B). An approximate $NO_3$ concentration of $3 \times 10^{12}$ molecule $cm^{-3}$ was generated via the 248 nm photolysis (R16) of $N_2O_5$ ($\approx 10^{15}$ molecule $cm^{-3}$) in all quenching experiments.



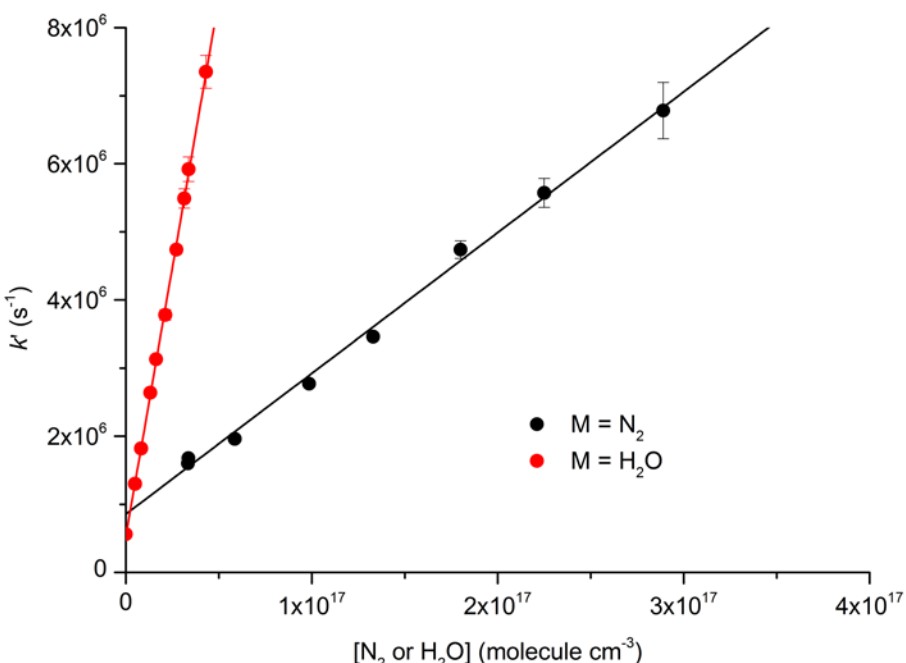

**Figure 6**: Plots for the determination of total quenching rate coefficients for $NO_3^*$ with $N_2$ (R10) and

with $H_2O$ (R12) at 296 K: $k_{10} = (2.1 \pm 0.1) \times 10^{-11}$ cm$^3$ molecule$^{-1}$ s$^{-1}$; $k_{12} = (1.6 \pm 0.3) \times 10^{-10}$ cm$^3$

molecule$^{-1}$ s$^{-1}$.




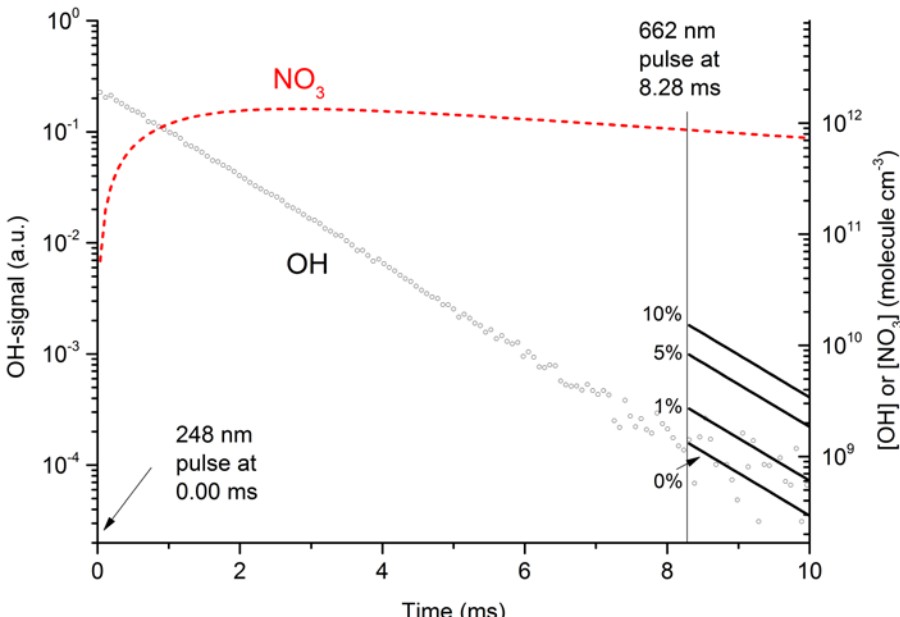

**Figure 7**: Plot of primary OH and expected OH (solid lines after 8.28 ms) from $NO_3$ * + $H_2O$ at

various values (0 to 10%) of $k_{12b} / k_{12}$. The initial OH concentration (right y-axis) was $2.0 \times 10^{12}$

molecule $cm^{-3}$. The dashed red line displays the calculated $NO_3$ concentration, which at 8.28 ms (time

of 662 nm excitation pulse) was $9 \times 10^{11}$ molecule $cm^{-3}$. In these conditions 50% of available $NO_3$ was

excited to $NO_3^*$ by absorption at 662 nm; 35% of this $NO_3^*$ proceeded to react with $H_2O$ in (R12),

with the balance quenched by $N_2$ or $HNO_3$. The solid lines ($t > 8.28$ ms) represent expected OH

signals for values of $k_{12b} / k_{12}$ between 0 and 10%.