# Peer review of "Reactive quenching of electronically excited $NO_2^*$ and $NO_3^*$ by $H_2O$ as potential sources of atmospheric HOx radicals"

_Atmospheric Chemistry and Physics, 2018_

## Referee Comment (RC1) · Anonymous Referee #1 · 8 Aug 2018

This manuscripts reports a careful lab study on the reactive quenching of electronically excited atmospheric NO2* and NO3* radicals by water vapour as potential primary sources of atmospheric HOx radicals.
The topic is relevant for ACP.

The manuscript is divided into two parts.
Firstly, the reactive quenching rate of NO2* by water vapour is re-investigated in order to elucidate the controversially discussed role of this photoreaction as a relevant primary source of atmospheric OH. From their results the authors inferred a new upper limit of $6 \times 10^{-6}$ (at an excitation wavelength of 562 nm) about an order of magnitude lower than previously measured. Extended measurements across the whole non-dissociative part of the absorption spectrum of NO2 lead to the conclusion that atmospheric OH formation by NO2* + H2O is insignificant.
Secondly, the reactive quenching of NO3* was studied. Although daytime concentration of NO3 are usually very low the authors estimated that under certain conditions the formation rate of OH by the title reaction could compete with the primary formation of OH from ozone photolysis. The measured quenching rate constant for the bath gas N2 compares well with the literature value of Nelson et al. The measured total quenching rate constant for H2O for which only one measurement is available in the literature was found to be a factor of four lower. In an elegant series of well-thought-out experiments it is shown that the majority of excited NO3* is deactivated by water vapour via radiationless processes and only an upper limit of <3% was found to react with water to form OH.

The experiments were conducted in a well-proven and tested apparatus. Concentrations of reactants are directly measured by UV/VIS absorption spectroscopy reducing the uncertainties to <10%. All relevant experimental details are reported, measurement uncertainties and experimental limitations are always well defined. Figures are meaningful and support the thorough and critical analysis and the discussion of the results.
I recommend the well written manuscript for publication once the following minor comments are addressed.

**Specific comments**

Is there a deeper reason for the application of two different OH formation pathways (R13, R14) to convert the LIF raw signal to OH concentrations? In the experimental section both methods are described but it is pointed out later in the text (page 7, line 9) that in the presence of NO2 193 nm photolysis (R14) would produce small amounts of $O^1D$ atoms from NO2 in a two-photon process which finally rules out this reaction for the NO2* study.
On the other hand Figure 3 shows OH formation from the photolysis of water vapour. Why not from HNO3 photolysis as applied in the NO2* experiment? Is there a corresponding figure?
This point certainly needs some clarification and I would suggest to re-write the first paragraph of page 6 accordingly.

In order to enhance the concentration of excited NO3* radicals to a maximum of 50% the absorption transition at 662 nm was "saturated" using very high laser power intensities.
How can saturation be experimentally verified and which laser fluence was applied?

The discussion of the potential OH formation rate from NO3* + H2O closes with the words "… it may still represent an important contribution to OH formation in some environments."
I am not happy with this general and more or less meaningless statement. It should be either substantiated or omitted.

**Technical corrections**

Page 3, line 30 – I calculate an excitation energy of 180.85 kJ/mol from the absorption of 662 nm photons. Hence, the given values of the reaction enthalpies require correction as well.

Page 5, line 25 – add type of light source, presumably *low pressure Hg lamp*

Page 7, line 5   – add absorption cross section of H2O at 193 nm
line 28 – typo: OH

Page 9, line 1   – typo: excited state
line 15 – typo: $1.6 \times 10^{-10}$   /  factor of four…

Page 12 – reference Dillon et al.: two names in the list of  co-authors are missing
(Vereecken, Peeters)

Page 17, Fig 3 – add concentration of H2O: $1.5 \times 10^{17}$ molec cm$^{-3}$
– correct 'the the'

---

## Referee Comment (RC2) · M. Blitz (Referee) · 10 Aug 2018

The authors have carried out a thorough and comprehensive study and showed that there is no evidence that NO2 excited by visible light can react with H2O to form OH + HONO. The present study has lowered the upper limit for OH formation and implies that this reaction has no atmospheric impact.

In addition, the possibility that the reaction between NO3 excited by visible light and H2O can form OH was investigated. No evidence for OH formation from this reaction was observed. However, the assigned upper OH yield for this reaction does not wholly rule it out from having some atmospheric impact. The other potential reactive channel

[Figure]

HO2 + HONO is discussed but not investigated.

This paper is fine for publication with just a few minor corrections.

---

## Referee Comment (RC3) · Anonymous Referee #3 · 20 Aug 2018

The authors present new data that help to better constrain reactions of excited state NO2* and NO3* with H2O. These are challenging experiments to conduct and interpret, and the authors have done a great job. This is a very interesting paper and should be published after my suggestions below have been addressed.

General comments

1 - Error bars /estimates.

Errors estimates should be added throughout the manuscript (especially to tables and graphs), if possible.

[Figure]

2 - Manuscript organization.

The results and discussion section gives (too many) experimental details. For example, pg 6, line 25, begins with "A Nd-YAG pumped dye-laser was used to generate 532 and 567 - 647 nm light ... Reagent concentrations and conditions for these experiments ..." These are experimental details. Consider some re-organization (shifting of text) to improve the organization of the paper.

3 - The authors expanded the wavelength regions over which the title reactions have been studied to above 532 nm and below 647 nm. It would be useful if the authors could comment on this chemistry occurring at lower wavelengths (in the intro and future work sections).

Specific comments

Pg 1 line 15 "which is". Please rephrase to avoid unnecessary confusion– is it $k_{12}$, or are $k_{10}$ and $k_{12}$, more than a factor of 3 smaller?

pg 1 line 20 "370 nm". This seems high (thinking of Talukdar et al., Geophys. Res. Lett., 25, 143-146, 10.1029/97gl03354, 1998), but perhaps I am not read up on the latest literature. Can you please provide a reference?

Pg 2 line 18 – strike comma after (1997)

Pg 3 line 29 "A simple analysis . . . " to pg 4 line 2 "(12c). Please provide more detail as to how these calculations were performed (I can guess but shouldn't have to) and a reference to the parameters going into these calculations.

Pg 4 line 12 "two" Figure 1 indicates that one SZA is 27 degrees. What's the other, and what was the result?

Pg 4 line 12 "TUV". More detail is needed. For example, please indicate TUV version, time of day, assumptions made albedo, aerosol optical depth, etc.

pg 4, line 13 "on average" I am not clear what is being averaged. Results at the two

[Figure]

SZA at noon? Results from 10 to 2 o'clock? Dusk to dawn?

Pg 5 lines 24 and 27. Please indicate at what wavelengths HNO3 and H2O concentrations were determined (or were both measured at 185 nm? If so, how were both determined simultaneously?)

Pg 7 line 17. "it is immediately apparent, that our data are not consistent with". Can you add error bars to make the line of reasoning more convincing?

Pg 10 line 13 – replace "is" with "are"

Pg 11 discussion of HO2+HONO pathway, line 22 "Given that our experiments were blind to formation of HO2 or HONO". This was very interesting to read, though speculative. It may be worth pointing that HO2 reacts with NO3, but (probably) too slowly to matter in these experiments.

Pg 20 Figure 5. There are data points between the "2" and "3" line, and it is not clear what data set they belong to. Can you color-code the odd and even data differently, perhaps?

---

## Author Comment (AC1) · 30 Aug 2018

The following text contains the reviewer's comments (black), our replies (blue) and the changes made to the manuscript (red).

| Reviewer 1 |
|---|
| This manuscripts reports a careful lab study on the reactive quenching of electronically excited atmospheric NO2* and NO3* radicals by water vapour as potential primary sources of atmospheric HOx radicals. The topic is relevant for ACP. |
| The manuscript is divided into two parts. Firstly, the reactive quenching rate of NO2* by water vapour is re-investigated in order to elucidate the controversially discussed role of this photoreaction as a relevant primary source of atmospheric OH. From their results the authors inferred a new upper limit of 6x10-6 (at an excitation wavelength of 562 nm) about an order of magnitude lower than previously measured. Extended measurements across the whole non-dissociative part of the absorption spectrum of NO2 lead to the conclusion that atmospheric OH formation by NO2* + H2O is insignificant. |
| Secondly, the reactive quenching of NO3* was studied. Although daytime concentration of NO3 are usually very low the authors estimated that under certain conditions the formation rate of OH by the title reaction could compete with the primary formation of OH from ozone photolysis. The measured quenching rate constant for the bath gas N2 compares well with the literature value of Nelson et al. The measured total quenching rate constant for H2O for which only one measurement is available in the literature was found to be a factor of four lower. In an elegant series of well-thought-out experiments it is shown that the majority of excited NO3* is deactivated by water vapour via radiationless processes and only an upper limit of <3% was found to react with water to form OH. |
| The experiments were conducted in a well-proven and tested apparatus. Concentrations of reactants are directly measured by UV/VIS absorption spectroscopy reducing the uncertainties to <10%. All relevant experimental details are reported, measurement uncertainties and experimental limitations are always well defined. Figures are meaningful and support the thorough and critical analysis and the discussion of the results. |
| I recommend the well written manuscript for publication once the following minor comments are addressed. |
| We thank the reviewer for this positive assessment of our manuscript. |
| Is there a deeper reason for the application of two different OH formation pathways (R13, R14) to convert the LIF raw signal to OH concentrations? In the experimental section both methods are described but it is pointed out later in the text (page 7, line 9) that in the presence of NO2 193 nm photolysis (R14) would produce small amounts of O1D atoms from NO2 in a two-photon process which finally rules out this reaction for the NO2* study. |
| On the other hand Figure 3 shows OH formation from the photolysis of water vapour. Why not from HNO3 photolysis as applied in the NO2* experiment? Is there a corresponding figure? |
| This point certainly needs some clarification and I would suggest to re-write the first paragraph of page 6 accordingly. |
| As evident in Table 1, both R13 and R14 can be (and were) used to calibrate the OH signal. The point we make is that use of 193 nm (H$_2$O photolysis, R14) can not be used in the presence of NO$_2$. In the first paragraph of page 6 we already explain this: |
| When using (R14) to calibrate the OH signal, the NO$_2$ supply to the experiment was replaced with N$_2$, and 193 nm light used to dissociate OH from the H$_2$O already present (in unchanged conditions of pressure, temperature and [H$_2$O]). |
| We have amended the caption to Figure 3 to remove any ambiguity: |
| The open circles are OH-calibrations obtained by the 193 nm photolysis of H$_2$O (in the |

absence of $NO_2$) at different laser fluences (mJ cm$^{-2}$).

In order to enhance the concentration of excited NO3* radicals to a maximum of 50% the absorption transition at 662 nm was "saturated" using very high laser power intensities. How can saturation be experimentally verified and which laser fluence was applied?

The laser fluences (in photons cm$^{-2}$ per pulse) are listed in Table 2. These were used to calculate, via the Beer-Lambert law, which fraction of $NO_3$ would have been promoted to the excited state. If this exceeded 50%, the transition was considered saturated. No attempt was made to very this experimentally.

The discussion of the potential OH formation rate from NO3* + H2O closes with the words "… it may still represent an important contribution to OH formation in some environments." I am not happy with this general and more or less meaningless statement. It should be either substantiated or omitted.

We have substantiated this statement:

…it is non-negligible compared to OH production rates from photolysis of $O_3$ (see section 1.2) and may still represent an important contribution to OH formation in environments where OH generation via traditional routes involving absorption of UV radiation is suppressed e.g. at high-latitudes in winter.

We recognise that this statement is still very qualitative, but feel that a more detailed analysis (requiring global-scale model calculation of OH formation) is not presently warranted.

Page 3, line 30 – I calculate an excitation energy of 180.85 kJ/mol from the absorption of 662 nm photons. Hence, the given values of the reaction enthalpies require correction as well.

The calculation of the reaction is infect correct even though the energy of 662 nm photon was wrongly listed. We have corrected the text and write:

Absorption of a 662 nm photon (the wavelength of maximum absorption by $NO_3$, see Figure 1), provides an excitation energy of $\approx$ 181 kJ mol$^{-1}$. Using compilations of enthalpies of formation (Wagman et al., 1982; Davis et al., 1993; Ruscic et al., 2004; Ruscic et al., 2005; Ruscic et al., 2006) we calculate that formation of radical products from $NO_3^*$ is exothermic: by 110 kJ mol$^{-1}$ for OH + $HNO_3$ (R12b) and by 81 kJ mol$^{-1}$ for $HO_2$ + HONO (R12c).

Page 5, line 25 – add type of light source, presumably *low pressure Hg lamp*

Information added. We now write:

…. Hg-line at 184.95 nm,

Page 7, line 5 – add absorption cross section of H2O at 193 nm

Addition made. We write:

The OH concentration was calculated using a 193 nm cross section for $H_2O$ of 2.1 $\times$ 10$^{-21}$ cm$^2$ molecule$^{-1}$ (Sander et al., 2011).

line 28 – typo: OH

Correction made

Page 9, line 1 – typo: excited state

Correction made

line 15 – typo: 1.6 x 10-10 / factor of four…

Correction made

Page 12 – reference Dillon et al.: two names in the list of co-authors are missing (Vereecken, Peeters)

Correction made

Page 17, Fig 3 – add concentration of H2O: 1.5x1017 molec cm-3 – correct 'the the'

Correction made

| **Reviewer 2** |
|---|
| The authors have carried out a thorough and comprehensive study and showed that there is no evidence that NO2 excited by visible light can react with H2O to form OH + HONO. The present study has lowered the upper limit for OH formation and implies that this reaction has no atmospheric impact. In addition, the possibility that the reaction between NO3 excited by visible light and H2O can form OH was investigated. No evidence for OH formation from this reaction was observed. However, the assigned upper OH yield for this reaction does not wholly rule it out from having some atmospheric impact. The other potential reactive channel HO2 + HONO is discussed but not investigated.

This paper is fine for publication with just a few minor corrections.

We thank the reviewer for this positive assessment of our manuscript. |

| **Reviewer 3** |
|---|
| The authors present new data that help to better constrain reactions of excited state NO2* and NO3* with H2O. These are challenging experiments to conduct and interpret, and the authors have done a great job. This is a very interesting paper and should be published after my suggestions below have been addressed.

We thank the reviewer for this positive assessment of our manuscript. |

| 1 - Error bars /estimates.
Errors estimates should be added throughout the manuscript (especially to tables and graphs), if possible

The only Figure which can usefully present error bars is Figure 6. This already has error bars. We now describe the source of these error bars in the caption:

The error bars are statistical uncertainty ($2\sigma$) from the fits to OH-decays as exemplified in Figure 5.

We feel we have adequately addressed sources of uncertainty when deriving (upper limits to) the yields we present. We write, for example:

The upper limits were calculated from the minimum observable OH-signal (assumed to be twice the RMS noise levels on the OH-signal) and accounting for uncertainty in parameters such as laser fluence (30%), $NO_2$ concentration (10%) and concentration of $H_2O$ (10%).
And
The major sources of uncertainty in the calculated OH-yield are uncertainty in the measurement of laser fluences (30%) required to calculate the initial OH and $NO_3^*$ concentrations and assumptions related to the (unmeasured) $NO_3$ time profile. |

| 2 - Manuscript organization.
The results and discussion section gives (too many) experimental details. For example, pg 6, line 25, begins with "A Nd-YAG pumped dye-laser was used to generate 532 and 567 - 647 nm light ... Reagent concentrations and conditions for these experiments ..." These are experimental details. Consider some re-organization (shifting of text) to improve the organization of the paper.

In the results and discussion section we have removed reference to the lasers
$NO_2$ was excited at a number of different wavelengths, 532 and 567 - 647 nm; reagent concentrations and conditions for these experiments are given in Table 1
As several different experiment type using different excitation wavelengths, calibration schemes and conditions were used, we prefer to keep some experimental details in the results and discussions section (i.e. close to the results being discussed) rather than moving it to the experimental section. |

| 3 - The authors expanded the wavelength regions over which the title reactions have been studied to above 532 nm and below 647 nm. It would be useful if the authors could comment on this chemistry occurring at lower wavelengths (in the intro and future work sections). |

We see no real benefit in giving a more detailed description of chemistry at shorter excitation wavelengths and prefer to keep the discussion focussed.
* * *
Pg 1 line 15 "which is". Please rephrase to avoid unnecessary confusion– is it k12, or are k10 and k12, more than a factor of 3 smaller?

We have clarified and now write:

Our value of $k_{12}$ is more than a factor of four smaller than one previously reported value.
* * *
pg 1 line 20 "370 nm". This seems high (thinking of Talukdar et al., Geophys. Res. Lett., 25, 143-146, 10.1029/97gl03354, 1998), but perhaps I am not read up on the latest literature. Can you please provide a reference?

Although the quantum yields are low, $O_3$ does indeed photolyse to generate $O(^1D)$ out to 370 nm. We now provide a reference:

Most atmospheric OH is believed to be generated via a combination of primary photolytic processes involving e.g. $O_3$ ($\lambda \leq 370$ nm, (IUPAC, 2018)) (R1, R2) and HONO ($\lambda$: 280-370 nm, (IUPAC, 2018)).
* * *
Pg 2 line 18 – strike comma after (1997)

Correction made
* * *
Pg 3 line 29 "A simple analysis . . . " to pg 4 line 2 "(12c). Please provide more detail as to how these calculations were performed (I can guess but shouldn't have to) and a reference to the parameters going into these calculations

In fact, the calculations (of reaction enthalpy) are described immediately below this sentence, which we have removed. We now give a citation for the enthalpies of formation used:

Absorption of a 662 nm photon (the wavelength of maximum absorption by $NO_3$, see Figure 1), provides an excitation energy of $\approx 181$ kJ mol$^{-1}$. Using compilations of enthalpies of formation (Wagman et al., 1982; Davis et al., 1993; Ruscic et al., 2004; Ruscic et al., 2005; Ruscic et al., 2006) we calculate that formation of radical products from $NO_3^*$ is exothermic: by 110 kJ mol$^{-1}$ for OH + $HNO_3$ (R12b) and by 81 kJ mol$^{-1}$ for $HO_2$ + HONO (R12c).
* * *
Pg 4 line 12 "two" Figure 1 indicates that one SZA is 27 degrees. What's the other, and what was the result?

This was misleading. We have modified the text and now refer only to 27°:

(calculated for 50 °N at a zenith angles of 27°, TUV)
* * *
Pg 4 line 12 "TUV". More detail is needed. For example, please indicate TUV version, time of day, assumptions made albedo, aerosol optical depth, etc.

We now provide this information:

(calculated using the TUV program (http://cprm.acom.ucar.edu/Models/TUV/Interactive_TUV/) for 50 °N at a zenith angle of 27°, overhead $O_3$ column of 300 Du, a surface albedo of 0.1 and an aerosol optical depth of 0.235).
* * *
pg 4, line 13 "on average" I am not clear what is being averaged. Results at the two C2 SZA at noon? Results from 10 to 2 o'clock? Dusk to dawn?

In fact we refer to averaging over the $NO_3$ absorption spectrum. We now write:

we calculate that, averaged over the $NO_3$ absorption spectrum.......
* * *
Pg 5 lines 24 and 27. Please indicate at what wavelengths HNO3 and H2O concentrations were determined (or were both measured at 185 nm? If so, how were both determined simultaneously?)

We have added the following sentence to clarify the procedure.

In experiments where both $HNO_3$ and $H_2O$ were present, they were added sequentially so first the optical density due to a single component was measured before the second was added and the resultant total optical density monitored.
* * *
Pg 7 line 17. "it is immediately apparent, that our data are not consistent with". Can you add error bars to make the line of reasoning more convincing?

As we present an upper limit, error bars on our data are not appropriate. The uncertainty is discussed in subsequent text. We now have removed "immediately apparent" and write:
Within experimental uncertainty (see below) our data are clearly not consistent with the large yield of OH reported by Li et al. (2008).

Pg 10 line 13 – replace "is" with "are"
Correction applied

Pg 11 discussion of HO2+HONO pathway, line 22 "Given that our experiments were blind to formation of HO2 or HONO". This was very interesting to read, though speculative. It may be worth pointing that HO2 reacts with NO3, but (probably) too slowly to matter in these experiments.
Yes, reaction between these two radicals would not be a source of OH.

Pg 20 Figure 5. There are data points between the "2" and "3" line, and it is not clear what data set they belong to. Can you color-code the odd and even data differently, perhaps?
We have now used different symbol types to separate the individual decays.